# Organisation and delivery of liaison psychiatry services in general hospitals in England: results of a national survey

Andrew Walker,[1] Jessica Rose Barrett,[2] William Lee,[3] Robert M West,[4] Elspeth Guthrie,[4] Peter Trigwell,[5] Alan Quirk,[6] Mike J Crawford,[6,7] Allan House[4]

For numbered affiliations see end of article.

**Correspondence to**
Dr Elspeth Guthrie;
e.a.guthrie@leeds.ac.uk

## ABSTRACT

**Objectives** To describe the current provision of hospital-based liaison psychiatry services in England, and to determine different models of liaison service that are currently operating in England.

**Design** Cross-sectional observational study comprising an electronic survey followed by targeted telephone interviews.

**Setting** All 179 acute hospitals with an emergency department in England.

**Participants** 168 hospitals that had a liaison psychiatry service completed an electronic survey. Telephone interviews were conducted for 57 hospitals that reported specialist liaison services additional to provision for acute care.

**Measures** Data included the location, service structures and staffing, working practices, relations with other mental health service providers, policies such as response times and funding. Model 2-based clustering was used to characterise the services. Telephone interviews identified the range of additional liaison psychiatry services provided.

**Results** Most hospitals (141, 79%) reported a 7-day service responding to acute referrals from the emergency department and wards. However, under half of hospitals had 24 hours access to the service (78, 44%). One-third of hospitals (57, 32%) provided non-acute liaison work including outpatient clinics and links to specialist hospital services. 156 hospitals (87%) had a multidisciplinary service including a psychiatrist and mental health nurses. We derived a four-cluster model of liaison psychiatry using variables resulting from the electronic survey; the salient features of clusters were staffing numbers, especially nursing; provision of rapid response 24 hours 7-day acute services; offering outpatient and other non-acute work, and containing age-specific teams for older adults.

**Conclusions** This is the most comprehensive study to date of liaison psychiatry in England and demonstrates the wide availability of such services nationally. Although all services provide an acute assessment function, there is no uniformity about hours of coverage or expectation of response times. Most services were better characterised by the model we developed than by current classification systems for liaison psychiatry.

## Strengths and limitations of this study

► A comprehensive national survey of liaison psychiatry services in acute hospitals, at a time of increased government investment, and debate about equity of access to mental and physical healthcare.
► The survey obtained 100% response rate for all hospitals in England with an emergency department.
► Classification of services was carried out using model-based clustering.
► A limitation was that service provision was reported by the services themselves rather than based on independent observation.
► Mental health services provided by clinicians outside of the liaison psychiatry service were not comprehensively reviewed.

teaching and research in non-psychiatric clinical settings: the 'liaison' referred to is therefore between psychiatry and other clinical disciplines. In the UK, it has been largely based in acute ('general') hospitals. The origins of liaison psychiatry can be traced to the 1930s but substantial growth only occurred in the postwar decades[1–4]: The UK's Royal College of Psychiatrists established its faculty of liaison psychiatry in 1997 and first published a competency-based curriculum for postgraduate training in liaison psychiatry in 2009.[5]

The case for liaison psychiatry services rested initially on observations that the prevalence of many psychiatric problems in acute hospitals is well above general population levels and that such comorbidities can pose particular management challenges.[6] People with problems such as psychosis, panic, delirium or self-harm may present to the emergency department (ED), or their difficulties may become apparent on inpatient wards—perhaps requiring rapid assessment and intervention.[7] Liaison psychiatry services also see people with more long-standing problems such as difficulty adjusting to severe physical illness,

## INTRODUCTION

Liaison psychiatry is the subspecialty of psychiatry concerned with clinical practice,

or complex physical health and mental health conditions—such work mostly being undertaken in outpatient clinics.

Recent interest in liaison psychiatry in the UK[8] has been focused on two issues—cost savings that might result from the service, and the need to provide equitable access to emergency care for all patients regardless of whether their problems are primarily physical, psychiatric or a combination.[9]

The suggestion that financial savings from timely psychiatric intervention are sufficient to pay for the liaison psychiatry service undertaking that intervention, the so-called 'cost-offset' effect, is not new.[10] Most recently it attracted interest in the UK following the publication of a report from one English hospital which reported that their 'Rapid Assessment Intervention and Discharge' (RAID) service achieved reductions in average inpatient lengths of stay in the target population of up to 4 days, even for patients not directly seen by the service.[11 12]

An important influence in current debate in the UK has been the classification of hospitals in terms of four service grades proposed by Aitken *et al*[13] (see online supplementary appendix for details). This classification was based on services already in existence that were capable of delivering certain levels of coverage in the hospital. It has been used to inform commissioning of services, with the aim that all hospitals with EDs should have a liaison service meeting such standards by 2020.[14]

Here, we present findings from the most detailed survey of liaison psychiatry services yet undertaken in England, describing staffing levels and their relation to other mental health services associated with the acute hospitals in which they are located. The aim was to describe the current provision of hospital-based liaison psychiatry services in England and to determine different models of liaison service that are currently operating.

This work arises from the first phase of a programme of research funded by the National Institute for Health Research, to evaluate the cost-effectiveness and efficiency of different configurations of liaison psychiatry services in England (Liaison Psychiatry-Measurement and EvaluationLP-MAESTRO) (http://www.nets.nihr.ac.uk/projects/hsdr/135808), and an annual mapping survey of liaison services funded by Health Education England, National Health Service (NHS) England and the Royal College of Psychiatrists. A prior survey of liaison psychiatry services was carried out in 2013, and this paper describes the second study,[15] which was carried out in conjunction with the LP-MAESTRO programme.

## METHOD
### Setting and sample
The sample consisted of all acute hospitals in England that had an ED at the time. Acute Trusts were identified from the NHS website (www.nhs.uk/servicedirectories/pages/nhstrustlisting.aspx) and individual hospitals were then identified from Trust websites.

Within each hospital liaison psychiatry service, we identified components of service—typically defined by the part of the hospital covered by that component—for example: ED, ward referrals, links to specialist services, liaison psychiatry outpatient clinics.

Each component of the service might then have different characteristics such as staff mix, working hours, performance targets, patient groups seen.

### Design
Cross-sectional two-stage survey conducted by email and telephone interview.

### Measures
The email survey ran between 14 May and 30 April 2015. The survey was brief and allowed flexible (free text) responses. Response was by email or telephone. Non-responding hospitals and missing response items were followed up by email and telephone. The questions asked in the email are given in online supplementary appendix 1.

We derived two variables describing RAID services. The first, 'original RAID', is based on the description provided in Tadros *et al*[12] of the service evaluated at Birmingham City Hospital; the second, 'modified RAID', is based on the profile of current services in Birmingham still known as RAID. We characterised each service according to whether they met the criteria for either of these service types. We also used responses on staffing level or working practice to classify each service according to recent guidance from NHS England that was created to help commissioners in planning service delivery.[13] The grades used in the guidance are as follows: Comprehensive (full liaison provision), Enhanced 24 (staffed according to the original RAID model), Core 24 (provides acute provision for a hospital with an ED, but no outpatient work) and Core (intended for less busy hospitals); services not meeting Core criteria were classified as subCore (see online supplementary appendices 2 and 3 for details).

A telephone interview survey ran between 16 July and 30 September 2015. It was undertaken to obtain further details about services that reported that they provided liaison services in addition to provision for acute care of patients in the ED or on the acute hospital wards (eg, outpatient services or specialist renal input).

Data from the survey have been published by the Royal College of Psychiatrist.[15] What we present in this paper is a reclassification of these data (carried out by the LP-MAESTRO team), a statistical analysis of the data using cluster analysis and results of the telephone survey, none of which has been previously published.

### Patient and participant involvement
Further work in the LP-MAESTRO programme will focus on patient experience. This will involve use of an on-line survey with service users (patients and carers) and non-psychiatric clinical staff who use liaison psychiatry services, with the aim of identifying additional outcomes

and aspects of service that are not well characterised by quantitative work.

The results of the work presented in this paper will be disseminated to the liaison teams at each of the hospitals who took part in the study interviews.

## ANALYSIS

The main analyses were undertaken with R statistical software V.3.2.2 (R Core Team 2016).

A latent class model[16] was fitted to perform clustering of responding hospitals. The number of clusters to be used was determined by minimising the Bayesian information criterion (BIC) because the BIC tends to favour less complexity. Models were fitted only if the number of observations (168) exceeded the number of parameters used in the model, thus ensuring a positive number of df. Other hospital properties were extracted from the survey and used as covariates in this model-based clustering approach.

Many of the variables used in clustering were categorical. Variables which might have been regarded as continuous were categorised so that all were handled in a similar way. For example, the number of hours of operation of the service was defined as three categories: 40–80 hours per week, 81–167 hours per week and 168 (=7×24) hours per week. Since all variables to be clustered were categorical, the polytomous latent class analysis package poLCA V.1.4.1 (16) with R statistical software V.3.2.0 (R core team 2015) was used for all analyses. The latent class function made use of the expectation–maximisation algorithm and there was the possibility of convergence to a local maximum rather than a global maximum. To overcome this, multiple starts were used.[17]

## RESULTS
### Staffing and working practices

Data were obtained on all 179 acute hospitals identified in England: 168 (94%) reported that they had a liaison psychiatry service; 11 had no service. All 168 hospitals with a liaison service completed the electronic survey and answered questions in follow-up emails and telephone calls, ensuring that there were no missing data.

Twelve services were nurse-only services. All other services were multidisciplinary and all included at least a psychiatrist of some grade and a mental health nurse. One hundred and forty-one hospitals (79%) reported at least one consultant psychiatrist as part of the team (total number=195), 95 hospitals (53%) reported other psychiatrists (non-consultant grade), 42 hospitals (23%) reported a psychologist or psychological therapist as part of the team, 26 hospitals (15%) reported allied health professionals and 52 hospitals (29%) reported other mental health staff. All 168 hospitals with a liaison service had nursing staff as part of the team and there were 1384 whole time equivalents working in liaison services at the time of the survey.

One hundred and forty-one hospitals (79%) provided a 7-day service and 15 hospitals (8%) provided a service Monday to Friday. Of the 141 hospitals process, 78 (55%) reported a 24 hours 7-day service.

Out of the 168 hospitals that had a liaison service, 75 hospitals (45%) had target response times of 1 hour or less for referrals from the ED and 73 hospitals (43%) reported target response times to referrals from the wards as less than 1 day. Sixty-four hospitals (38%) had no target response time.

Nearly all of the liaison services (99%) saw patients who were referred following self-harm (167 hospitals) and many saw patients for assessment of alcohol and substance misuse (106 hospitals 63%). Only 37 services (22%) saw patients with learning disabilities. Forty-four services (26%) had separate older adult and working-age adult teams.

Fifty-seven hospitals (34%) reported a service or component of service that did more than serve the acute care pathway, and 4 of these hospitals operated virtually separate liaison services for acute and non-acute referrals.

### Classification according to RAID and Core

Of the 168 hospitals, only 8 met the original RAID criteria and 35 met criteria for modified RAID. Ten liaison services had the term RAID in their title, without meeting either of the RAID criteria.

Of the 168 hospitals, 1 was rated as Comprehensive, 3 were Enhanced24 (2%), 13 were Core24 (8%), 18 were Core (11%) and 133 were subCore (79%). The Comprehensive rated service met modified RAID criteria, two of the Enhanced24 services met Original RAID criteria and the final Enhanced24 service did not meet either RAID criterion. Of those services that met either RAID criterion, 28/41 (68%) were rated as Core or subCore.

### Types of liaison psychiatry service: results from cluster analysis

We used data from the email survey to cluster the services using characteristics listed in table 1.

The minimum value of BIC was with four clusters, but the value for three clusters was very near the minimum also. Hence, a decision was required between three and four clusters and we decided the model with four clusters was more interpretable and useful. Hospitals were assigned to a cluster according to their modal probability, that is, hospitals were labelled as a certain cluster when the model gave a probability of membership of that cluster to be larger than that of any other. Table 2 shows the modal cluster membership tabulated against hospital characteristics.

Model-based clustering identified four classes. These do not represent discrete categories but rather services that are relatively similar to each other in a diverse landscape.

► Cluster 1: Services tended to be based in smaller hospitals, had the smallest numbers of consultant staff and nurses. Only a minority offered 24 hours 7-day cover, few had predefined response times and none

**Table 1** The characteristics derived from survey responses used to distinguish liaison psychiatry services in the model-based clustering

| | |
|---|---|
| Labelling | 1. Does the name of the service include 'RAID'? |
| | 2. Is the service classified as subCore, Core or does it meet one of the definitions: Core24, Enhanced or Comprehensive? |
| | 3. Does the service operate 7 days per week or for less than 7 days? |
| | 4. How many hours per week is the service provided? |
| Coverage | 5. Does the service claim to cover all mental health? |
| | 6. Is there a dedicated working-age adults (18–65) team? |
| | 7. Is there a dedicated older adults (65+) team? |
| Work done | 8. Does the service undertake work from the emergency department? |
| | 9. Does the service undertake in-reach work? |
| | 10. Does the service operate an outpatient clinic? |
| | 11. Does the service have pathways other than acute pathways? |
| | 12. What is the response time for the emergency department? |
| | 13. What is the response time for the wards? |
| Other aspects of hospitals (Note that for some variables, the value assigned may have been inferred from other survey responses rather than taken from the direct response given) | 14. No of services within a hospital (1, 2 or 3) |
| | 15. No of providers of services (1 or 2) |
| | 16. No of hospital beds |
| | 17. No of nurses employed by the liaison psychiatry service |
| | 18. No of consultants |
| | 19. No of services |
| | 20. No of service providers |
| Additional variables | 21. Does the service meet the original RAID criteria? |
| | 22. Does the service meet the modified RAID criteria? |

RAID, Rapid Assessment Intervention and Discharge.

met either of the RAID criteria. Few offered outpatient clinics and none offered care outside the acute pathway.

► Cluster 2: Services were most likely to meet one of the RAID criteria, providing 24 hours 7-day cover, working to response time targets for ED and ward referrals and concentrating exclusively on the acute care pathway with no follow-up outpatient clinics.

► Cluster 3: Services were more diverse with some offering 24 hours 7-day services, but the defining feature, was that they also offered outpatient clinics and covered care outside the acute care pathway; they had the highest number of consultants and nurses—number of nurses being an important determinant of the probability of membership in this cluster.

► Cluster 4: These were also diverse services, one-third offering outpatient clinics and work outside the acute pathway; only a minority provided 24 hours 7-day cover or worked to response time targets and none met either of the RAID criteria. All these hospitals had separate teams for working-age adults and for older persons.

### The nature of clinical services: telephone interviews
We undertook telephone interviews covering 57 hospitals and 61 separate liaison services; four hospitals had two distinctly different liaison teams. The telephone interviews reflected the clustering with most of those interviewed being in clusters 3 and 4: cluster 1: n=8, cluster 2: n=4, cluster 3: n=30 and cluster 4: n=19.

### ED referrals
Fifty-seven out of the 61 services (93%) saw acute referrals from the ED. Most (53, 87%) were available Monday to Sunday, and n=32 (52%) were available 24 hours a day. Forty-nine (80%) of the services responded to referrals of adult patients of any age, but entry criteria could be quite specific—for example, one service saw all working-age adults throughout the day and older age adults only for the first half of the night. Most ED liaison psychiatry teams (51, 84%) were multidisciplinary although five consisted of nursing staff only.

Referrals from the ED varied considerably in scope and numbers; out of the 46 reported referral rates, the mean number of weekly referrals was 36 (minimum=1, maximum=100). In addition to assessment, most services that were interviewed (36, 59%) offered ED patients outpatient follow-up.

All of the 57 services which served the ED had key performance indicators, and almost three in four (n=43, 70%) measured patient outcome in some way.

### Ward referrals
Fifty-seven services accepted ward referrals. Most (n=46, 75%) were available to wards 7 days, and nearly one-third (n=20, 33%) were available to wards 24 hours a day. Almost three out of four (n=44, 72%) responded to referrals from wards for adult patients of any age, five services responded to referrals from wards only for older adults and nine responded to referrals from wards only for working-age adults.

**Table 2** Hospital characteristics according to cluster membership

| Hospital characteristic | Cluster 1 n=46, (%) | Cluster 2 n=35, (%) | Cluster 3 n=43, (%) | Cluster 4 n=44, (%) |
|---|---|---|---|---|
| RAID | | | | |
| Name has RAID in title | 1 (2) | 7 (20) | 10 (23) | 1 (2) |
| Not codable for RAID | 3 (6) | 1 (3) | 5 (12) | 1 (2) |
| Not RAID | 43 (94) | 11 (31) | 20 (46) | 43 (98) |
| Original RAID | 0 | 0 | 6 (14) | 0 |
| Modified RAID | 0 | 23 (66) | 12 (28) | 0 |
| Core classification | | | | |
| SubCore | 45 (98) | 23 (66) | 29 (67) | 36 (82) |
| Core | 1 (2) | 5 (14) | 7 (16) | 5 (11) |
| Core24 | 0 | 7 (20) | 3 (7) | 3 (7) |
| Enhanced24 | 0 | 0 | 3 (7) | 0 |
| Comprehensive | 0 | 0 | 1 (2) | 0 |
| Service operates 7 days | 33 (72) | 34 (97) | 34 (79) | 40 (91) |
| Hours of operation | | | | |
| 40–80 hours | 15 (33) | 1 (3) | 8 (19) | 15 (34) |
| 81–167 hours | 16 (34) | 7 (20) | 13 (30) | 15 (34) |
| 7×24=168 hours | 15 (33) | 27 (77) | 22 (51) | 14 (31) |
| Serves all MH | 32 (70) | 21 (60) | 26 (60) | 19 (43) |
| Dedicated WAA | 0 | 0 | 0 | 44 (100) |
| Dedicated OAA | 0 | 0 | 0 | 44 (100) |
| OP clinic | 2 | 0 | 43 (100) | 14 (32) |
| Non-acute pathway | 0 | 0 | 43 (100) | 14 (32) |
| Response time to the ED | | | | |
| <1 hour | 3 (6) | 35 (100) | 25 (58) | 12 (27) |
| 1.5–4 hours | 17 (37) | 0 | 4 (9) | 11 (25) |
| Not stated or >4 hours | 26 (57) | 0 | 14 (33) | 21 (48) |
| Response time to wards | | | | |
| <24 hours | 8 (17) | 29 (83) | 23 (53) | 13 (30) |
| 36 hours to 5 days | 8 (17) | 6 (17) | 0 | 8 (18) |
| Not stated | 30 (65) | 0 | 20 (46) | 23 (52) |
| Hospital beds | | | | |
| 50–447 | 19 (41) | 14 (40) | 12 (28) | 8 (18) |
| 447–621 | 20 (43) | 8 (24) | 16 (37) | 16 (36) |
| 622–1943 | 7 (15) | 13 (54) | 17 (40) | 20 (45) |
| No of FTE nurses | | | | |
| 0.5–6.0 | 25 (54) | 8 (23) | 11 (26) | 16 (36) |
| 6.1–9.4 | 15 (33) | 11 (31) | 11 (26) | 15 (34) |
| 9.5–24.0 | 6 (13) | 16 | 21 (49) | 13 (30) |
| No of FTE consultants | | | | |
| 0.0–0.5 | 27 (57) | 7 (20) | 9 (21) | 15 (34) |
| 0.6–1.4 | 16 (35) | 13 (37) | 12 (28) | 13 (30) |
| 1.5–9.2 | 3 (6) | 15 (43) | 22 (51) | 16 (36) |

**Table 2** Continued

| Hospital characteristic | Cluster 1 n=46, (%) | Cluster 2 n=35, (%) | Cluster 3 n=43, (%) | Cluster 4 n=44, (%) |
|---|---|---|---|---|
| Single service provider | 45 (98) | 33 (94) | 41 (95) | 40 (91) |
| Two service providers | 1 (2) | 2 (6) | 2 (5) | 4 (9) |
| No of services within same hospital | | | | |
| 1 | 44 (96) | 35 (100) | 39 (91) | 27 (61) |
| 2 | 1 (2) | 0 | 4 (9) | 16 (36) |
| 3 | 1 (2) | 0 | 0 | 1 (2) |

ED, emergency department; FTE, full-time equivalent; MH, mental health; OP, outpatient; RAID, Rapid Assessment Intervention and Discharge; WAA, working-age adults; OAA, older adults service.

Again most ward teams (53, 87%) were multidisciplinary teams although four consisted of medical staff only, three consisted of nursing staff only and one staffed the ward team with psychiatrists and psychologists. Based on 47 reported referral rates the mean number of weekly referrals was 25. All responding services assessed patients and offered short-term follow-up on medical wards, and over half (n=34, 56%) offered outpatient follow-up.

Fifty of those we interviewed had key performance indicators for wards, and 42 (69%) measured patient outcome in some way for ward referrals.

### Self-harm referrals
All but two services accepted referrals for self-harm. Fifty-one (84%) offered a 7-day service, 29 (48%) provided a service 24 hours a day and most (n=46, 75%) offered a service to adults of all ages.

Most services we interviewed (54, 89%) said their self-harm teams were multidisciplinary. All services assessed patients on wards and approximately half (n=31, 51%) offered short-term follow-up on medical wards. Only eight services described a separate self-harm outpatient clinic, although several services described seeing small numbers of selected patients.

### Liaison psychiatry with named specialist services
Twenty services (33%) provided specialist liaison services to at least one named specialist service or department in the hospital. A total of 31 different specialist services were reported; the most frequently reported were gastroenterology (n=5), hepatology (n=4) palliative care (n=4), maternity, neurology, trauma and transplant (n=3 each).

### Outpatient clinics
Thirty-three services (54%) provided a general liaison psychiatry outpatient clinic and 28 services (46%) reported running an outpatient clinic for particular specialist groups; 20 services had both types of clinic. Twenty-two of the general liaison psychiatry clinics saw patients of any adult age and 11 clinics saw working-age adults only. Thirteen clinics were staffed with a multidisciplinary team, 15 were solely medical, 2 had nursing staff only and 3 clinics included a psychologist. Referrals to the general liaison psychiatry clinic came predominantly from the acute hospital in which they were based (n=26). Thirty-one clinics offered short-term treatment and follow-up and 18 offered longer term treatment and follow-up.

The most commonly reported specialist clinics were for medically unexplained symptoms (n=7), diabetes (n=4), bariatric surgery patients (n=4), respiratory disease (n=3) and perinatal psychiatry (n=3). Most clinics offered some form of psychological therapy—problem-solving therapy, (n=46, 75%) motivational interviewing, (n=38, 62%) cognitive–behavioural therapy (n=35, 57%) behavioural activation (n=31, 51%) or interpersonal therapy (n=26, 43%).

### Other acute hospital mental health providers
In order of frequency, 47 liaison services (77%) coexisted in the acute hospital with separate drug or alcohol services, 44 (72%) with clinical psychology and 22 (36%) coexisted with health psychology. We identified a wide range of other services—for particular patient groups or for overlapping patient groups by other agencies.

### Referral to local service providers
The ease of referral to other mental health services for patients requiring follow-up was also investigated. All services said they could routinely refer to a local community mental health team, 52 (85%) could refer to a crisis team routinely, 54 (89%) could refer routinely to drug and or alcohol and the same number to older adult psychiatry. Over half (n=36, 59%) of services could routinely refer to clinical psychology and 19 (31%) to health psychology.

### Non-clinical activity
All services we interviewed provided some form of non-clinical work in the form of staff training or educational sessions, medico-legal assessments, advice to managers and others. The most common non-clinical services were: dementia training (n=17); research and service evaluation (n=16); organisational support and advice to acute hospital staff (n=15); delirium training (n=11); mental capacity act training (n=10).

### DISCUSSION
We identified widespread availability of liaison psychiatry services in acute hospitals in England. Liaison psychiatry

teams were customarily multidisciplinary and most services saw all acute mental health problems in the hospital and adults of all ages. Our findings suggest that there has been a gradual but continued expansion in liaison psychiatry services over the last 20 years, as evidenced by several previous surveys including those focusing on consultant posts in the British Isles,[18 19] services in a particular area of England[20] and the one previous unpublished national survey undertaken at the request of NHS-England (LPSE-1). As an example of the expansion, the number of consultant posts in liaison psychiatry in the British Isles more than doubled from 43 in 1998[18] to 93 in 2003.[19] The findings of the current survey suggest a further increase with 195 consultant posts in liaison psychiatry in England alone.

We found 11 hospitals that reported having no liaison service at all, which is concerning given that one of the targets set by NHS England in the Five Year Forward View for Mental Health is that by 2020 no acute hospitals should be without all-age mental health liaison services in EDs and inpatient wards.[21]

Only one-third of services offered outpatient clinics and non-acute care. The range of such activities was wide, with more than 30 different specialist services. There was however very little in common between services about which specialist activities were supported. Surprisingly few services (just over 10% of our total sample) reported running clinics that supported longer term follow-up and treatment opportunities, a sine qua non for the management of problems with living with long-term illness or of severe and chronic medically unexplained symptoms.[22] This gap in service provision was most striking in self-harm services; we found that the majority of services offered acute assessment but no service offered routine therapeutic treatment for service users.

Very few of the services we surveyed readily fitted into the current commissioning framework,[13] or the RAID framework, so the classification into Core, Core24, Enhanced24 or Comprehensive had limited value in discriminating between hospitals, and neither descriptive framework proved useful in identifying those services that reported 24 hours 7-day acute services.

For these reasons we sought a more practical, data-driven approach to describing service types. We chose model-based clustering to do so. Alternative approaches would have been to use one of many heuristic algorithms such as hierarchical clustering, k-means, self-organising maps, graph-theoretic approaches or support vector machines. A generative mixture model has the advantage that it can be more general and provides a statistical framework within which to decide on the number of clusters present. Model-based clustering has also been found to perform better than other approaches in identifying clusters.[16] The models were simply parameterised since there were only 168 observations. Within a diverse picture of provision, our cluster analysis did reveal some patterns of service—the three most obvious features that distinguished between services were the hours of cover

and response time standards, the likelihood of providing non-acute care in outpatients and the decision to have separate teams for older and working-age adults. Size of hospital and staffing levels (especially nursing) were important associations with the type of service offered. This suggests that when services scale up from the basic provision represented by cluster 1 (and found in smaller hospitals) they do so in one of these three directions—increasing intensity of acute work, developing outpatient and non-acute work or developing specialist old age teams.

Our findings have implications for those commissioning and those providing services.

First, we found widespread availability of liaison psychiatry services in English acute hospitals, but most teams were poorly resourced compared with published recommendations. Second, whatever local decisions are made about liaison psychiatry, our survey suggests national coordination of services is lacking. Third, we were struck by the unexpectedly low levels of longer term outpatient treatment provision. Problems of adjustment to long-term illness, persistently poor adherence to challenging treatment regimens, medically unexplained symptoms and severe somatoform disorders all form part of the raison d'etre for liaison psychiatry and their management requires sustained professional input, in the hospital as well as in community settings. Our results confirm previous findings about the low national level of provision for people who harm themselves.

There are several limitations to our study. Our approach to surveying provided a rather general high-level account of services that does not do full justice to the richness and diversity of provision in multicomponent services. Reliance on a single (or occasionally a second) informant at each stage may have led to missing or inaccurate information. The service descriptors we used were based on self-report, and we have not verified them with direct independent observation. Our sampling strategy meant we did not collect information on specialist hospitals without EDs, so we did not collect data on rare but important facilities in specialist hospitals. Our survey was entirely hospital focused and while we are aware of (and involved in) initiatives to develop and evaluate primary care-based liaison psychiatry services, they were not studied here.

There is increasing interest in the idea that well-run liaison psychiatry services can be both important in improving quality of care in acute hospitals and cost-effective.[23–25] UK liaison services are changing rapidly, with a round of investment especially in provision for emergency assessment and response.[26] A further national survey of all English acute hospitals has been completed and the results will be published in the near future. It is hoped this latest survey will provide further coverage of a rapidly changing landscape.

Liaison psychiatry services in the UK are being encouraged by their specialty representative group in the Royal College of Psychiatrists to use a standardised package of outcome measures, the Framework for Routine Outcome

Measurement in Liaison Psychiatry,[27] to enable benchmarking against national norms. On the basis of these and other evaluation exercises, we expect to achieve an increasingly detailed and nuanced account of the nature and impact of liaison psychiatry—a subspecialty that has a valuable role in providing genuinely coordinated and inclusive healthcare.

**Author affiliations**
¹Clinical Research Network National Coordinating Centre, National Institute of Health Research Clinical Research Network, Leeds, UK
²National Collaborating Centre for Mental Health, Royal College of Psychiatrists, London, UK
³Institute of Translational and Stratified Medicine, Plymouth University Peninsula Schools of Medicine and Dentistry, Plymouth, UK
⁴Leeds Institute of Health Sciences, University of Leeds, Leeds, UK
⁵National Inpatient Centre for Psychological Medicine, Leeds and York Partnership NHS Foundation Trust, Leeds, UK
⁶College Centre for Quality Improvement, Royal College of Psychiatrists, London, UK
⁷Centre for Psychiatry, Department of Medicine, Imperial College London, London, UK

**Acknowledgements** This work was supported by the efforts of S Keane, M Heneghan, S Chee, J Lewis, S Jayakumar and H Lucas at the Royal College of Psychiatrists and the NIHR Clinical Research Network (CRN) Yorkshire and Humber. J Hewison, C Czoski Murry, C Smith, C Hulme, S Tubeuf, A Martin, S Relton from the Leeds Institute of Health Sciences and Matthew Fossey from the Centre for Mental Health Research are fellow collaborators on the programme.

**Contributors** AH, MJC, EG and PT conceived of the research. JRB and WL carried out the survey. AAW coordinated the telephone interviews. RMW carried out the statistical analysis. AQ and all other authors contributed to the manuscript and approved the final version.

**Funding** The present study arises from two funding initiatives: a national survey of staffing and structure in liaison psychiatry services in acute hospitals completed on behalf of the Royal College of Psychiatrists and the National Collaborating Centre for Mental Health commissioned by NHS England (Liaison Psychiatry Survey of England 2015, LPSE 2015) and a research study funded by the UK's National Institute for Health Research HS&DR programme to evaluate the effectiveness and cost- effectiveness of liaison psychiatry services (LP-MAESTRO project number 13/58/08).

**Disclaimer** The views and opinions expressed therein are those of the authors and do not necessarily reflect those of the HS&DR, NIHR, NHS or the Department of Health.

**Competing interests** None declared.

**Patient consent** Not required.

**Ethics approval** NHS Ethical permission (REC reference: 15/NS/0025) and Trust level approval was obtained for the telephone interviews.

**Provenance and peer review** Not commissioned; externally peer reviewed.

**Data sharing statement** Survey data are available from the author WL.

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
