## [Reviewer comments · BMJ Open]

ARTICLE DETAILS

TITLE (PROVISIONAL)	The Organisation and Delivery of Liaison Psychiatry Services in General Hospitals in England: results of a National Survey
AUTHORS	Walker, Andrew; Barrett, Jessica; Lee, William; West, Robert; Guthrie, Elspeth; Trigwell, Peter; Quirk, Alan; Crawford, Mike; House, Allan

VERSION 1 – REVIEW

REVIEWER	A/Prof Rob Parker NT Top End Mental; Health Services, Australia
REVIEW RETURNED	05-Apr-2018

GENERAL COMMENTS	Good Paper that will add to the literature on C/L Psychiatry. The authors may wish to check the accuracy of the dates described in Line 47 of the Method: Measures section as they appear to be going backwards in time!!
---

REVIEWER	Dr Jim Bolton South West London & St George's Mental Health NHS Trust and St George's University of London, UK
REVIEW RETURNED	05-Apr-2018

GENERAL COMMENTS	The publication of this research is timely in view of the national recommendation that all English acute hospitals with an emergency department should have a liaison psychiatry service by 2020. This paper will provide a valuable contribution to both national and local discussions about service provision and models of care. As the authors note, the subject of the study also pertains to the national's stated political aim of ensuring equitable access to both mental and physical health services. I found the paper to be thorough, appropriately referenced and well-written and I have no suggestions for amendment.
--

REVIEWER	Philip R. Muskin, MD, MA Columbia University Medical Center, United States
REVIEW RETURNED	10-May-2018

GENERAL COMMENTS	Excellent study that shows the diversity of liaison services. It clearly outlines the need for enhanced support to provide more services everywhere in the country.
---

VERSION 1 – AUTHOR RESPONSE

Many thanks for reviewing this manuscript. We have made the following changes as requested by the reviewers and editor.

We have checked and revised the dates from 14 May 2015 and 30 April 2015 to the 14 May 2014 to April 2015. Highlighted in tracked changes, page 5.

We have removed the appendices and uploaded them separately in a pdf format.

We have added specific mention of AQ and his role as co-author.

We have added ref for unpublished work referred to in the introduction.

We have uploaded a clean and marked copy of the main document.

We have uploaded a STROBE checklist

We have provided a brief description of the degree of patient involvement in the design and conduct of the study, as well as plans to disseminate the results to study participants.

We have removed the 'unpublished' citation as the source 'Lee W. A National Survey of Liaison Psychiatry in England. 2013' already given.

We hope these changes are satisfactory.